# Expression and Role of IL-1β Signaling in Chondrocytes Associated with Retinoid Signaling during Fracture Healing

**DOI:** 10.3390/ijms21072365

**Published:** 2020-03-29

**Authors:** Tsuyoshi Shimo, Hiroaki Takebe, Tatsuo Okui, Yuki Kunisada, Soichiro Ibaragi, Kyoichi Obata, Naito Kurio, Karnoon Shamsoon, Saki Fujii, Akihiro Hosoya, Kazuharu Irie, Akira Sasaki, Masahiro Iwamoto

**Affiliations:** 1Division of Reconstructive Surgery for Oral and Maxillofacial Region, Department of Human Biology and Pathophysiology, School of Dentistry, Health Sciences University of Hokkaido, Hokkaido 061-0293, Japan; fujii@hoku-iryo-u.ac.jp; 2Division of Histology, Department of Oral Growth and Development, School of Dentistry, Health Sciences University of Hokkaido, Hokkaido 061-0293, Japan; takebeh@hoku-iryo-u.ac.jp (H.T.); hosoya@hoku-iryo-u.ac.jp (A.H.); irie@hoku-iryo-u.ac.jp (K.I.); 3Departments of Oral and Maxillofacial Surgery, Okayama University Graduate School of Medicine, Dentistry, and Pharmaceutical Sciences, Okayama 700-8525, Japan; pphz1rke@okayama-u.ac.jp (T.O.); ykunisada@okayama-u.ac.jp (Y.K.); sibaragi@md.okayama-u.ac.jp (S.I.); dental1818@yahoo.co.jp (K.O.); aksasaki@md.okayama-u.ac.jp (A.S.); 4Department of Oral Surgery, Tokushima University Graduate School, Tokushima 770-8504, Japan; kurio.naito@tokushima-u.ac.jp; 5Division of Clinical Cariology and Endodontology, Department of Oral Rehabilitation, School of Dentistry, University of Hokkaido, School of Dentistry, Hokkaido 061-0293, Japan; karnoon@hoku-iryo-u.ac.jp; 6Department of Orthopaedics, University of Maryland School of Medicine, Baltimore, MD 21201, USA; Masahiro.Iwamoto@som.umaryland.edu

**Keywords:** fracture healing, chondrocyte, interreukin-1β, retinoid signaling

## Abstract

The process of fracture healing consists of an inflammatory reaction and cartilage and bone tissue reconstruction. The inflammatory cytokine interleukin-1β (IL-1β) signal is an important major factor in fracture healing, whereas its relevance to retinoid receptor (an RAR inverse agonist, which promotes endochondral bone formation) remains unclear. Herein, we investigated the expressions of IL-1β and retinoic acid receptor gamma (RARγ) in a rat fracture model and the effects of IL-1β in the presence of one of several RAR inverse agonists on chondrocytes. An immunohistochemical analysis revealed that IL-1β and RARγ were expressed in chondrocytes at the fracture site in the rat ribs on day 7 post-fracture. In chondrogenic ATDC5 cells, IL-1β decreases the levels of aggrecan and type II collagen but significantly increased the metalloproteinase-13 (Mmp13) mRNA by real-time reverse transcription-polymerase chain reaction (RT-PCR) analysis. An RAR inverse agonist (AGN194310) inhibited IL-1β-stimulated Mmp13 and Ccn2 mRNA in a dose-dependent manner. Phosphorylated extracellular signal regulated-kinases (pERK1/2) and p-p38 mitogen-activated protein kinase (MAPK) were increased time-dependently by IL-1β treatment, and the IL-1β-induced p-p38 MAPK was inhibited by AGN194310. Experimental p38 inhibition led to a drop in the IL-1β-stimulated expressions of Mmp13 and Ccn2 mRNA. MMP13, CCN2, and p-p38 MAPK were expressed in hypertrophic chondrocytes near the invaded vascular endothelial cells. As a whole, these results point to role of the IL-1β via p38 MAPK as important signaling in the regulation of the endochondral bone formation in fracture healing, and to the actions of RAR inverse agonists as potentially relevant modulators of this process.

## 1. Introduction

Fracture healing consists of three major stages: a reaction stage, a repair stage, and a remodeling stage [1]. After the fracture, the repair process begins with a hematoma and an inflammatory response at the fracture site. In the inflammatory phase, a lack of peripheral vasculature causes an anoxic environment that leads to the formation of a cartilage template, which initiates the differentiation process of endochondral bone formation. Endochondral bone formation is an indispensable process during the healing phase of fracture healing, beginning with the differentiation of bone marrow stem cells into chondrocytes, and then chondrocyte proliferation, differentiation, maturation, apoptosis, and vascular invasion [1].

The interaction of a tissue undergoing endochondral bone formation with inflammatory signals is a crucial process to be investigated. Most in vitro studies assessed the effects of inflammatory cytokines on either the osteogenic or chondrogenic differentiation capacity of mesenchymal stromal cells (MSCs). In particular, stimulation with interleukin 1beta (IL-1β) has been shown to enhance both the extent of mineralization and the expression of osteoblast-related genes during the culture of MSCs in osteogenic medium [2], and to inhibit MSC chondrogenesis in a dose-dependent manner [3]. IL-1β enhanced calcium deposition and the expression of bone morphogenetic protein (BMP)-2 mRNA by differential activations of Nuclear factor-κB(NF-κB) and ERK signaling in osteogenic and higher productions of matrix metallopeptidase (MMP)-13 in cultured bone marrow MSCs [4]. IL-1β inhibits the proliferation and differentiation of chondrocytes through a SOX-9-mediated mechanism [5]. The expression pattern of IL-1β during endochondral fracture repair is bimodal, peaking during the initial inflammatory phase and again during the later remodeling phase [6]. However, little is known about the exact nature of the inflammatory response in chondrocytes.

Retinoic acid (RA), the main active metabolite of retinol, is one of the key regulators of skeletal patterning during embryogenesis, and it regulates cartilage and bone development and growth [7]. RA is an important regulator of differentiation and cell proliferation in skeletal development, and retinoic acid receptor gamma (RARγ) has been demonstrated to be expressed specifically in growth plates from the chondrocyte proliferation zone to the hypertrophy zone [8]. Retinoids exert their effects by modulations of gene expression by two distinct classes of nuclear receptors, i.e., the retinoic acid receptors (RARα, -β, and -γ) and the retinoid X receptors (RXRα, -β, and -γ). An analysis of the effects of single and double RAR gene ablations in a murine model revealed compelling evidence that RARs are required for skeletal growth, matrix homeostasis, and growth plate function in postnatal mouse [7]. Overall, the specificity and the magnitude of RA action are controlled by the temporo-spatial patterns of the expression of RARs and RXRs and the endogenous cellular levels of RA [9,10]. Uchibe et al. reported that genetic or pharmacological interference with RARγ stimulated endochondral bone formation [11], and by using RARγ-null mice they observed that the absence of RARγ leads to more vigorous cartilage formation in response to bone defect. They also reported that RARγ antagonists stimulated cartilage formation, representing the early phase of endochondral bone formation and resulting in increased bone formation at later stages. In another study, RARγ mRNA was detected specifically in chondrocytes throughout the cartilaginous skeletal elements, and the RAR inverse agonist AGN194310 suppressed the RARγ agonist AGN204647-stimulated maturation-related genes expression in ATDC5 cells [12].

However, the effects of RAR inverse agonist effectors on chondrocytes and the mechanisms underlying the signaling action on IL-1β (a potent inflammatory cytokine that is upregulated during the fracture healing processes) have been largely unclear. The data that we obtained in the present study provide novel insights regarding the effect of an RAR inverse agonist on the regulation of IL-1β-induced chondrocyte-specific genes.

## 2. Results

### 2.1. Localization of IL-1β and RARγ in Fractured Rat Ribs

We first evaluated the expressions of IL-1β and RARγ in chondrocytes in a rat bone fracture model, making observations at day 7 after rib fracture. Toluidine blue staining of chondrocytes on day 7 sections is shown in Figure 1A,B. The immunohistochemical analysis revealed that IL-1β was distributed largely in chondrocytes and the surrounded mesenchymal cells in the bone marrow cells (Figure 1C,D), and that RARγ was expressed in chondrocytes at the fracture site in the ribs of mice on day 7 after fracture (Figure 1E,F).

### 2.2. Modulation of the Expression of Chondrocyte-Specific Genes’ mRNA by IL-1β and Retinoid Receptor Agonist in ATDC5 Cells

We next compared the effects of IL-1β on genes involved in endochondral bone formation with the effects of the RAR agonist on these genes described below. The aggrecan (Agr), the most abundant proteoglycan in the cartilage matrix, and type II collagen (Col II) for the markers of the chondrocytes, type X collagen (Col10), transglutaminase-2 (Tg2), Ccn2, and Mmp-13 for the markers of hypertrophic chondrocytes. Runx2 and Sox9 are the master transcription factors for bone and cartilage development respectively. RUNX2 inhibit chondrocyte differentiation and Sox9 is essential for cell survival in differentiated chondrocyte lineage cells. Confluent ATDC5 cells were switched to low-serum-containing medium and treated with 10 ng/mL IL-1β, 100 nM RARα agonist AGN195183, 100 nM RARγ agonist AGN204647, or 100 nM all-trans-retinoic acid (*ATRA*) for 8, 24, or 48 h and were then subjected to the real-time reverse transcription-polymerase chain reaction (RT-PCR) analysis. The results revealed that IL-1β was not closely involved in gene expression changes, and as shown in Figure 2A–C,E,F, aggrecan (Agr) (Figure 2A, 24 h: * *p* < 0.05), type II collagen (Col II) (Figure 2B), type X collagen (Col10) (Figure 2C), Sox9 (Figure 2E), and Runx2 (Figure 2F) showed a tendency to slightly decrease the mRNA expression. On the other hand, IL-1β was observed to increase the Mmp-13 mRNA expression 34-fold after 8 h (** *p* < 0.01) and 12-fold after 24 h (** *p* < 0.01) (Figure 2H). The Ccn2 mRNA showed a tendency to increase every 24 and 48 h after the addition of IL-1β (Figure 2D). The expression of Tg2 did not change after the addition of IL-1β (Figure 2G). The RARα agonist AGN195183 had no effect on any gene (Figure 2A–H). The RARγ agonist AGN204647 decreased the expression of Agr mRNA at 24 h (* *p* < 0.05) and 48 h (* *p* < 0.05) (Figure 2A) and increased the expression of the hypertrophic specific genes Col10 (Figure 2C, 24 h: ** *p* < 0.01, 48 h: * *p* < 0.05), Ccn2 (Figure 2D, 8 h: * *p* < 0.05, 24 and 48 h: ** *p* < 0.01), Tg2 (Figure 2G, 8, 24 and 48 h: ** *p* < 0.01) mRNA at 24 and 48 h (Figure 2C,D,G). ATRA increased the expression of Tg2 for 8 h (** *p* < 0.01) and 24 and 48 h (* *p* < 0.05) (Figure 2G). However, ATRA may be less specific inductive capacity than AGN204647 for ATDC5 cells the gene expression because of the pan-agonist of all the RARs [12].

### 2.3. The Effect of the RAR Inverse Agonist AGN194310 on IL-1β Modulated Genes

It has been reported that the RAR inverse agonist AGN194310 had little effect on RARα agonists but significantly suppressed the hypertrophic chondrocyte-specific gene expression induced by RARγ agonists in ATDC5 cells [12]. In other words, RAR inverse agonists have an inhibitory effect on RARγ signaling that is nearly specific to chondrocytes. We next investigated the effects of AGN194310 on genes involved in endochondral bone formation that is regulated by 10 ng/mL IL-1β. We observed that 1000 nM AGN194310 significantly reversed the down-regulations of Ccn2 (** *p* < 0.01) and Mmp13 (* *p* < 0.05) mRNA that were upregulated by 10 ng/mL IL-1β (Ccn2: ** *p* < 0.01, Mmp13: * *p* < 0.05) after 24-h treatment (Figure 3D,H). In addition, 100 nM AGN194310 had no effect on the expressions of Agr and Col10 mRNA which were downregulated by 10 ng/mL IL-1β (Agr: ** *p* < 0.01, Col10: ** *p* < 0.01) after 24-h treatment (Figure 3A,C). The expressions of Sox9 was not affected by IL-1β treatment, but Sox9 mRNA was significantly inhibited by the treatment of 10, 100, and 1000 nM AGN194310 (** *p* < 0.01, Figure 3E). The expressions of Col II, Runx2 and Tg2 mRNA were not affected by the 24-h treatment with IL-1β with or without the indicated amount of 100 nM AGN194310 (Figure 3B,F,G). The evidence from the ATDC5 cell cultures showed that the inhibition of endogenous RARγ by an RAR inverse agonist reduced the IL-1β-induced expressions of Mmp13 and Ccn2, indicating that retinoid signaling is a key regulator of the expressions of Mmp13 and Ccn2 in chondrocytes and may be an important downstream effector that is regulated by IL-1β signaling.

### 2.4. Analysis of the Actions of the RAR Inverse Agonist AGN194310 on IL-1β-Induced Mitogen-Activated Protein Kinase (MAPK) 

In the present study, to investigate whether the RAR inverse agonist AGN194310 treatment modulated the IL-1β activation of mitogen-activated protein kinase (MAPK), we treated cell cultures with 10 ng/mL IL-1β for different lengths of time with or without 100 nM AGN194310, and the cultures were processed for an immunoblot analysis of the phosphorylated levels versus total levels of the ERK1/2 and p38 MAPK. As shown in Figure 4A,B, the expressions of ERK1/2 and p38 MAPK started to increase after the IL-1β treatment. The ERK1/2 activation reached a plateau at 15 min, whereas the p38 MAPK activation reached a maximum at 24 h (Figure 4B, ** *p* < 0.01). AGN194310 at 100 nM inhibited IL-1β induced p38 MAPK activation from 15 min, and the inhibition was observed even at 24 h (Figure 4B, * *p* < 0.05). However, pERK1/2 showed a tendency to be stimulated by the treatment with AGN194310 from 15min and reached maximum for 30min compared with the IL-1β treatment (Figure 4B). To clarify the effect of AGN194310 on the phosphorylation of ATF2 through IL-1β signaling, we treated cells with 10 ng/mL IL-1β with or without 100 nM AGN194310, and confirmed the activation of ATF2 by conducting a p38 MAPK activity assay. The p38 MAPK activity was up-regulated by IL-1β treatment (Figure 4C, * *p* < 0.05), and the upregulation was inhibited by AGN194310 (Figure 4C, * *p* < 0.05).

### 2.5. Analysis of the Inhibition of Mitogen-Activated Protein Kinase (MAPK) on IL-1β Modulated Genes

Next, to determine whether ERK1/2 and p38 MAPK were involved in IL-1β’s action on the mRNA expression of aggrecan, Col2, Col10, Ccn2, Tg2, and Mmp13, we treated ATDC5 cells with 10 ng/mL IL-1β with or without 20 μM ERK1/2 kinase inhibitor PD98059 or p38 MAPK SB203580 for 24 h. The aggrecan, Col2, Col10, and Tg2 mRNA levels were not significantly changed by the treatment with IL-1β with or without PD98059 and SB203580 for 8 and 24 h (Figure 5A–C,E). Of note, the upregulation of Ccn2 and Mmp-13 mRNA induced by the IL-1β treatment were significantly inhibited by the SB203580 treatment as the control for 24 h (Figure 5D, * *p* < 0.05) and 8 h (Figure 5F, * *p* < 0.05) individually. However, 24 h after 10 ng/mL IL-1β treatment, the induction of Mmp-13 by IL-1β decreased by approx. 3-fold, and we observed that there was almost no effect of SB203580 (Figure 5F). PD98059 affected the neither IL-1β-induced Ccn2 nor Mmp13 expression (Figure 5D,F).

### 2.6. The Localization of MMP-13, CCN2, p-p38 MAPK, and α-SMA in Fractured Rat Ribs 

To provide a detailed characterization of Mmp-13, Ccn2, and activated p38 MAPK, we analyzed the expressions of Mmp-13, Ccn2, and p-p38 MAPK in the endochondral ossification on day 7 after a rib fracture. MMP13, CCN2, and p-p38 MAPK are expressed in hypertrophic chondrocytes (Figure 6A–C). A large number of α-smooth muscle actin (α-SMA)-positive blood vessels accumulated around the cartilage at 7 days after the fracture (Figure 6D), suggesting roles of MMP13 and CCN2 produced from chondrocytes as angiogenic factors and in extracellular matrix (ECM) remodeling. 

## 3. Discussion

Fracture healing is related to the endochondral ossification seen during embryonic development, but an important difference is the presence of an inflammatory phase during the bone-healing step. When a fracture occurs, angiogenesis and the local destruction of soft tissue form a hematoma that serves as a future template for callus formation, as shown in Figure 1 [13,14].

IL-1β is expressed in immune cells in the bone marrow in the early fractured process. In light of our present findings, it is not clear which immune cells are IL-1β-producing cells on day 7 after fracture, but there may be neutrophils and macrophages in a cartilaginous callus [15]. Immune cells are recruited to the fractured site’s secreted growth factors and cytokines, and they help mesenchymal cell recruitment; the mesenchymal progenitor cells are recruited to the damaged site and undergo chondrogenic differentiation [15]. The important role of IL-1β seems to involve the secretion from neutrophils to attract monocytes, which will differentiate to macrophages [16]. Although the role of IL-1β in immune cells in the early fracture process has been reported, there are few reports on the cartilage-callus stage of fracture healing.

The typical expression pattern of IL-1β in fracture healing is said to be bimodal, and we focused on the first peak that started from 24 h and ended after 7 days post-fracture [17,18]. The immunostaining demonstrated that the expression of IL-1β in the local area on the 7th day post-fracture was maintained, and it indicated that osteochondro-progenitors may differentiate into chondrocytes or osteoblasts accompanied by angiogenesis at the fracture site. A prior study showed that endochondral ossification leading to the formation of a cartilage callus and intramembranous ossification leading to the formation of periosteal callus occur, but the most important features of this process are the remodeling and ossification of cartilage callus [19]. IL-1β inhibited the proliferation and mineralizing potential of murine bone marrow stem cells, but it promoted the proliferation and mineralizing potential of preosteoblasts [18]. The relationship between the IL-1β signal acting in autocrine and paracrine manners through the endochondral pathway and the RAR signal involved in cartilage differentiation is not established, and it is an important process to be investigated.

The current data indicate that IL-1β and RARγ signals maintain appropriate levels of their expression in chondrocytes after a fracture and contribute to the transition from hypertrophic cartilage to bone remodeling. In the endochondral bone formation process in long bones, the expression of RARγ signal regulates the expressions of Mmp13 and Ccn2 mRNA, which have important roles in the resorption of hypertrophic cartilage from growth plates and the remodeling of newly deposited trabecular bone during long-bone development [12]. A general deletion of RARγ but not RARα in mice caused bone marrow defects characterized by hematopoietic stem cell alteration and even a marked reduction in trabecular bone during aging [20,21].

It is thus clear that there are intimate cross-talk interactions between the hypertrophic growth plate and bone marrow that are essential for the physiological progression of skeletal growth. Our present data indicate that RARγ signaling may bring about appropriate expression levels and the actions of these important hypertrophic chondrocyte-specific downstream effectors, contributing to a seamless transition from hypertrophic cartilage to trabecular bone. A general deletion of RARγ but not RARα in mice caused bone marrow defects characterized by altered hematopoietic stem cells and markedly reduced trabecular bone during aging, demonstrating that RARγ signaling is important in chondrocytes during endochondral bone formation.

The ERK1/2 and p38MAPK signaling pathways are involved in the chondrogenic differentiation program from the embryonic stage to postnatal development [22]. Activation of the p38 MAPK signaling pathway enhanced the proliferation of osteoblasts and the stimulation of osteoblast differentiation by the activation of RUNX2 by p38 MAPK [23,24]. In addition, activation of the p38 MAPK signaling pathway was shown to enhance the osteoblast proliferation and differentiation in a murine fracture model [25]. The p38MAPK signaling pathway in chondrocytes is activated by an RARγ agonist and induces the expressions of Mmp-13 and Ccn2 mRNA [12]. The Mmp-13expression induced by IL-1β was suppressed by an anti-inflammatory drug through p38 signaling pathways in chondrocytes, consequently abolishing the degradation of collagen II [26].

Our present findings suggest that the activation of p38 in response to IL-1β in chondrocytes leads to the activation of ATF2, and that IL-1β-mediated p38 activation regulates the proliferation of proliferating chondrocytes. ATF2 is expressed in resting and proliferating chondrocytes in the development of endochondral bone, and Atf2-deficient mice display uniform dwarfism due to reduced chondrocyte proliferation and disorganized chondrocyte columns [27]. On the other hand, it has been confirmed that p38 MAPK is also expressed in hypertrophic chondrocytes and that it promotes differentiation [28,29]. Therefore, p38 signaling by IL-1β may exert different functions depending on the stage of chondrocyte development; in particular, the target transcription factor ATF2 acts on resting/proliferating chondrocytes, and MEF2C and RUNX2 act on prehypertrophic/hypertrophic chondrocytes [27,30,31].

We don’t have data on the effect of AGN194310 on the activation of NF-κB involved in IL-1β signaling. ATRA reduced NF-κB activity and DNA binding [32], and p38 MAPK and NF-kB signaling have opposite biologic functions during inflammatory and osteogenic processes [33]. AGN194310 might regulate chondrocyte proliferation and differentiation to maintain endochondral ossification through accession of NF-κB and reduction of p38 MAPK signaling pathway [34]. However, the mechanisms of crosstalk between p38 MAPK and NF-kB signaling are not fully clarified [35], and require further investigation of AGN194310 effects on NF-kB signaling in future studies.

MMP13-deficient mice have been shown to delay fracture repair, suggesting a role for MMP-13 in the process of angiogenesis during fracture healing [36]. MMP-13 has been shown to contribute directly to angiogenesis, suggesting a physiological role for MMP-13 associated with cartilage and bone resorption in collagen remodeling during the angiogenesis process [37]. The importance of CCN2 in skeletogenesis was confirmed in CCN2-KO mice that exhibited multiple skeletal dysmorphisms as a result of impaired growth plate chondrogenesis and angiogenesis [38]. CCN2 is a potent angiogenic factor that directly stimulates endothelial cells’ proliferation and migration and ECM remodeling [39,40].

The vascular invasion of the cartilage callus is an important step during fracture healing [41], and fracture healing is significantly delayed when the vasculature is disrupted [42]. Hu et al. suggested that the vasculature may have a signaling role in regulating the transformation of chondrocytes to osteoblasts, and paracrine factors secreted from the vascular endothelial cells may trigger the chondrocyte-to-osteoblast transformation by activating the pluripotent stem cell programs, initiating cell division and/or stimulating the bone phenotype [43]. The immature cartilage in the fracture callus is avascular and the chondrocytes do not express vascular endothelial growth factor (VEGF); as the chondrocytes mature and become hypertrophic, they express VEGF and stimulate vascular invasion [43]. IL-1 induces VEGF production in chondrocytes through a distinct MAPK pathway [44]. p38 MAPK pathway mediates actin-based motility by regulating actin remodeling and cell contractility in response to VEGF in endothelial cells [45]. In our study, p38 MAPK activation was seen in the vascular endothelial cells near the hypertrophic chondrocytes after day 7 fracture (Figure 6C). The p38 pathway may be a major signaling pathway in the endothelial compartment as p38 MAPK plays central roles in regulating endothelial cell functions in response to oxidative stress [46]. This is illustrated histologically in the fracture callus by the localization of expression of IL-1β and the effect of an RAR inverse agonist in the chondrocytes, followed by ECM remodeling and the invasion of vascular endothelial cells into the transition zone (Figure 7).

## 4. Materials and Methods

### 4.1. Experimental Model for Fracture Healing

The right eighth rib of 26-week-old male Sprague–Dawley rats (Hokudo, Sapporo, Japan) were fractured as described [47]. Briefly, the rat was anesthetized, and the eighth rib on the right side was exposed and cut vertical to the axis with scissors. The experimental protocols were approved by the Ethics Review Committee for Animal Experimentation of the Health Sciences University of Hokkaido (The ethical permission code and permission date: 19-045, 29 March 2019)

### 4.2. Immunohistochemistry

For immunohistochemistry, dehydrated sections of rib fracture were treated with 0.3% H_2_O_2_ in phosphate-buffered saline (PBS; pH 7.4) for 30 min at room temperature to inactivate endogenous peroxidase. Sections were pretreated with 3% bovine serum albumin (BSA) in PBS for 30 min at room temperature, followed by incubation with primary antibodies against IL-1β (1:100, ab9722, Abcam, Cambridge, MA, USA), RARγ (1:100, #8965S, Cell Signaling Technology, Danvers, MA, USA), MMP-13 (1:100, ab39012, Abcam), CCN2 (1:100, ab6992, Abcam), α-SMA (1:1000, M0851, DakoCytomation, Glostrup, Denmark), and p-p38 (1:100, sc166182, Santa Cruz Biotechnology, Santa Cruz, CA, USA) overnight at 4 °C. Sections were reacted with Histofine Simple Stain rat MAX-PO (Multi; Nichirei, Tokyo) for 1 h at room temperature. Color was developed with the use of a liquid diaminobenzidine substrate-chromogen system (Dako, Carpinteria, CA, USA). Immunostained sections were then counterstained with methylene green.

### 4.3. Cell Cultures

A murine chondrogenic cell line, ATDC5, was purchased from the RIKEN Cell Bank (Tsukuba Science City, Japan). ATDC5 cells were cultured at a density of 1 × 10^4^ cells/cm^2^ in a 1:1 mixture of Dulbecco’s modified Eagle’s medium (DMEM) and Ham’s F12 medium (Gibco/BRL, Gaithersburg, MD, USA) containing 5% fetal bovine serum (FBS; Hyclone, Logan, UT, USA), followed by replacement with DMEM/F12 containing 5% FBS, 10 μg/mL human recombinant insulin (Wako Pure Chemical, Osaka, Japan), 10 μg/mL transferrin (Roche Diagnostics, Mannheim, Germany) and 3 × 10^−8^ M sodium selenite (Sigma) for the promotion of cell differentiation. The cells were then cultured at 37 °C for different periods up to 12 days under 5% CO_2_. RNA was extracted from the cultured ATDC5 cells when they became confluent (4 days after plating) and was then extracted every 2 days after confluence.

Day-10 cultures of ATDC5 cells were treated with IL-1β (10 ng/mL, R&D Systems, Minneapolis, MN, USA), 100 nM all-trans-retinoic acid (ATRA; Sigma, St. Louis, MO, USA), the 100 nM RARγ selective agonist AGN204647 [7], the 100 nM RARα-selective agonist AGN195183 [48], the 100 nM RAR inverse agonist AGN194310 [7], the selective inhibitor of ERK1/2 kinase PD98059 (20 μM, Calbiochem, La Jolla, CA, USA), the selective inhibitor of p38 kinase SB203580 (20 μM, Calbiochem), or combinations of these agents for 24 h.

### 4.4. Immunoblot Analysis

We conducted an immunoblot analysis to determine the levels of mitogen-activated protein kinase (MAPK) and activated MAPK by using cell lysates from the experimental cultures. The cells were lysed in an ice-cold lysis buffer (50 mM Tris-HCl, pH 7.4, containing 150 mM NaCl, 1% Triton X-100, 1% NP-40, 10 mM NaF, 100 mM leupeptin, 2 mg/mL aprotinin, and 1 mM phenylmethyl sulfonyl fluoride). The lysates were centrifuged at 16,000× *g* for 20 min at 4 °C, and the protein concentrations in the supernatant were determined by a bicinchoninic acid (BCA) assay. A 50-μg sample of each lysate was subjected to 12% sodium dodecyl sulfate-polyacrylamide gel electrophoresis (SDS-PAGE). The proteins were transferred to nylon membranes (Immobilon-P, Millipore, Bedford, MA, USA). The membrane was incubated with primary and secondary antibodies according to the ECL chemiluminescence protocol (RPN2109; Amersham Biosciences, Buckinghamshire, UK) to detect secondary antibody binding. Anti-ERK1/2 (rabbit IgG), pERK1/2 (Thy202/Tyr204, rabbit IgG), p38 (rabbit IgG), p-p38 (Tyr180/Tyr182, rabbit IgG) were purchased from Cell Signaling Technology Inc. (Danvers, MA, USA). Primary antibodies were used at a 1:200 dilution, and horseradish peroxidase-conjugated goat anti-rabbit antibodies (Cell Signaling Technology) were used as the secondary antibodies at a 1:1000 dilution. ImageJ software (NIH, Bethesda, MD, USA) was used to quantify band intensities on Immunoblots and the multiple comparison statistical analysis was performed.

### 4.5. p38 MAPK Assay

Cell extracts were incubated overnight with immobilized p38 MAPK (Thr180/Tyr182) monoclonal antibody (mAb) (#9219 Cell Signaling Technology). A kinase reaction was performed in the presence of 100 µM of cold ATP (Cell Signaling Technology) and 2 µg of ATF2 fusion protein (#9224S Cell Signaling Technology). The phosphorylation of ATF2 at Thr71 was measured by western blotting using phospho-ATF2 (Thr71) antibody (#9221S Cell Signaling Technology).

### 4.6. Real-Time Reverse Transcription-Polymerase Chain Reaction (RT-PCR)

Total RNA was isolated from ATDC5 cells by using TRIZOL reagent (Life Technologies, Rockville, MD, USA) according to the manufacturer’s recommendations. Complementary DNA was generated from 1 μg of total RNA in a final volume of 20 μL by using a First Strand cDNA Synthesis Kit (Invitrogen Corporation, Grand Island, NY, USA). The real-time RT-PCR was performed with a Light Cycler (Roche Molecular Biochemicals, Mannheim, Germany) in Light Cycler capillaries with a commercially available master mix containing Taq DNA polymerase and SYBR-Green I deoxyribonucleoside triphosphate (Light Cycler DNA Master SYBR-Green I; Roche Molecular Biochemicals). The primers used were the following: 5′-TGAGAGAGGCGAATGGAACG-3′ (forward) and 5′-TTCTGCCCGAGGGTTCTAGC-3′ (reverse) for aggrecan (Agr); 5′-AAGAACAGCATCGCCTACCT-3′ (forward) and 5′-CTTACCAGTGTGTTTCGTGC-3′ (reverse) for Col2; 5′-CATCAAGACGGAGCAGCTGAG-3′ (forward) and 5′- ATGGTCAGCGTAGTCGTATTG-3′ (reverse) for Sox9; 5′-TGATGACACTGCCACCTCTG-3′ (forward) and 5′-GAGGGATGAAATGCTTGGGA-3′ (reverse) for Runx2; 5′-TCTGTACAATAGGCAGCAGC-3′ (forward) and 5′-TAGGCGTGCCGTTCTTATAC-3′ (reverse) for Col10; 5′-CCAATGACAATACCTTCTGC-3′ (forward) and 5′-GAAAGCTCAAACTTGACAGG-3′ (reverse) for Ccn2; 5′-TTCCGAGTATGACT-3′ (forward) and 5′-GCCAATATCAGTCGGGAACA-3′ (reverse) for TG-2; 5′-CCAGACTATGGACAAAGATT-3′ (forward) and 5′-ATGCGATTACTCCAGATACT-3′ (reverse) for Mmp13; and 5′-TGAACGGGAAGCTCACTGG-3′ (forward) and 5′-TCCACCACCCTGTTGCTGTA-3′ (reverse) for glyceraldehyde-3-phosphate dehydrogenase (GAPDH). After the addition of primers (final concentration: 10 μM), MgCl_2_ (3 mM), and template DNA to the master mix, the real-time PCR were as follows: 40 cycles of 95 °C for 30 s, 60 °C for 30 s, and 72 °C for 30 s. Melting curves were obtained, RT-PCR was performed on the Light Cycler, and the data was analyzed by using ΔΔCT methods as we described previously [49]. 

### 4.7. Statistical Analysis

The data were analyzed using the unpaired Student’s t-test for the comparisons of two groups and Tukey’s test, Dunnett’s test and Bonferroni correction for the analysis of multiple group comparisons. Results are expressed as the mean ± standard deviation (SD). Probability (*p*)-values * *p* < 0.05 and ** *p* < 0.01 were considered significant.

## 5. Conclusions

IL-1 plays a critical role in bone fracture healing as the initial responder but also in the endochondral bone formation in chondrocytes, ECM remodeling, invasion of the vasculature, and the initiation of the cascades of the p38 MAPK signal to recruit cells to carry out the repair processes that are co-regulated with RAR signal. However, further studies are needed with the post-transcriptional gene regulation and in vivo studies to clarify the role of RAR inverse agonists on the fracture healing.

## Figures and Tables

**Figure 1 ijms-21-02365-f001:**
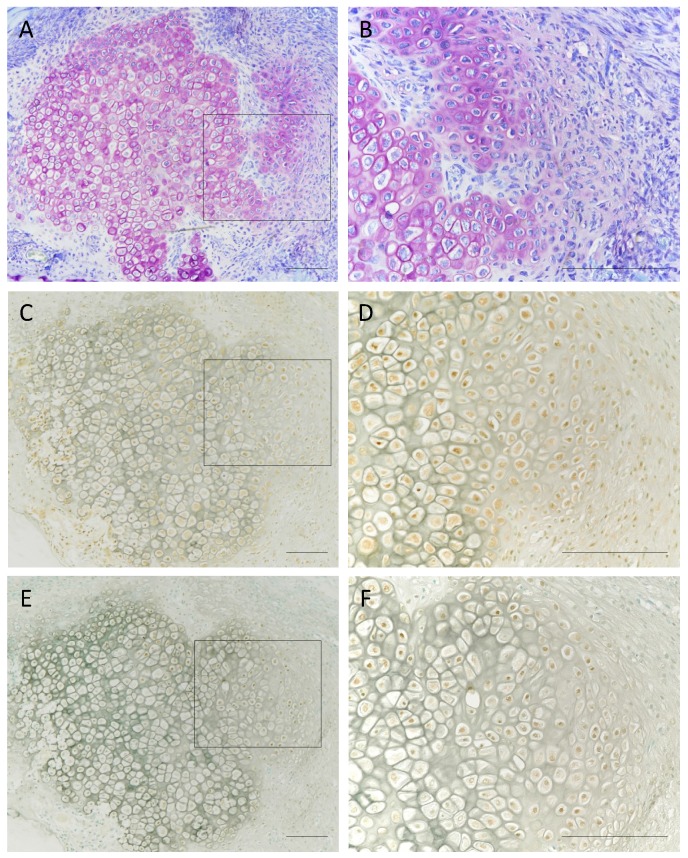
Histological appearance and localization of interleukin-1β (IL-1β) and retinoic acid receptor gamma (RARγ) in fractured rat ribs at 7 days after fracturing. Sections stained with Toluidine blue (**A**,**B**), IL-1β (**C**,**D**), and RARγ (**E**,**F**) illustrate endochondral ossification at day 7 post-fracture. Each right photo showing a histological section is a magnification of the rectangle-delimited area in the corresponding left photo. Bar, 100 μm.

**Figure 2 ijms-21-02365-f002:**
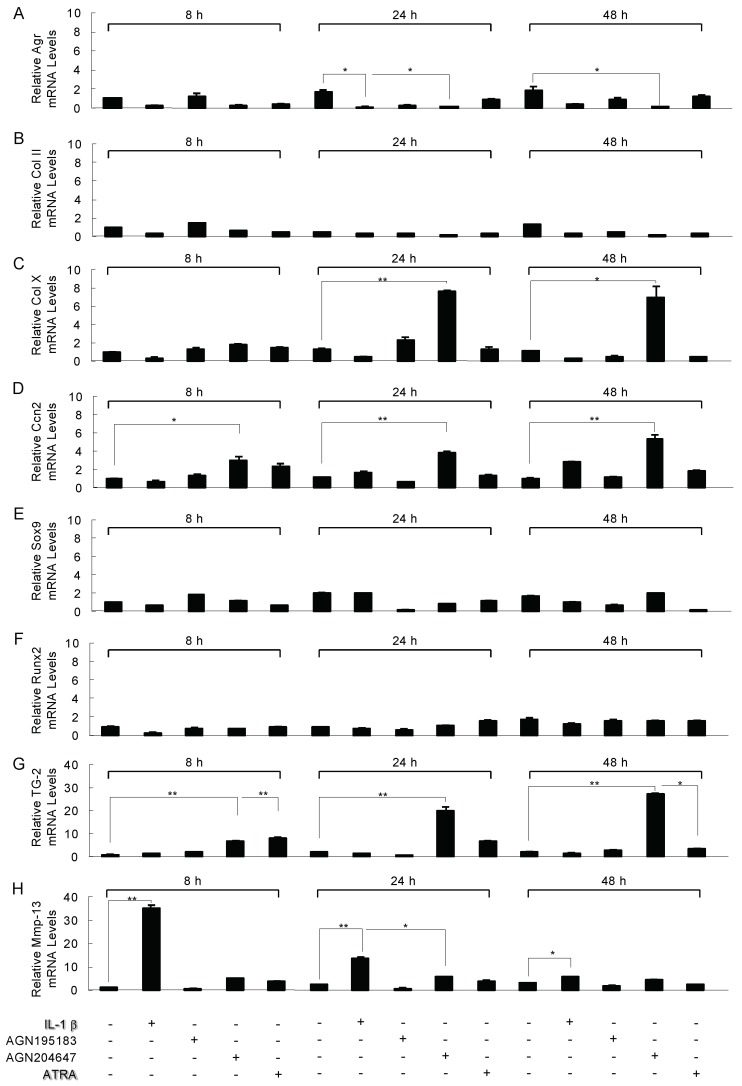
Modulation of the expression of chondrocyte-specific genes’ mRNA by IL-1β, retinoid receptor (RAR)-α agonist (AGN195183), RAR-γ agonist (AGN204647), or RAR inverse agonist (AGN194310) in 10-day cultures of ATDC5 cells. ATDC5 cells were exposed to 10 ng/mL IL-1β, 100 nM AGN195183, 100 nM AGN204647, or 100 nM all-trans-retinoic acid (ATRA) for 8, 24, or 48 h. Total RNA from these cells was used in a real-time RT-PCR for the analysis of the mRNA expressions of aggrecan (Agr) (**A**), type II collagen (Col II) (**B**), type X collagen (Col10) (**C**), Ccn2 (**D**), Sox9 (**E**), Runx2 (**F**), transglutaminase-2 (Tg2) (**G**), and matrix metalloproteinase-13 (Mmp13) (**H**). The values in the graph indicate the relative mRNA level of the fold changes. Columns, mean of triplicate determinations; Bars show mean ± SD. Statistically significant differences (* *p* < 0.05, ** *p* < 0.01) between the indicated groups are marked by asterisks.

**Figure 3 ijms-21-02365-f003:**
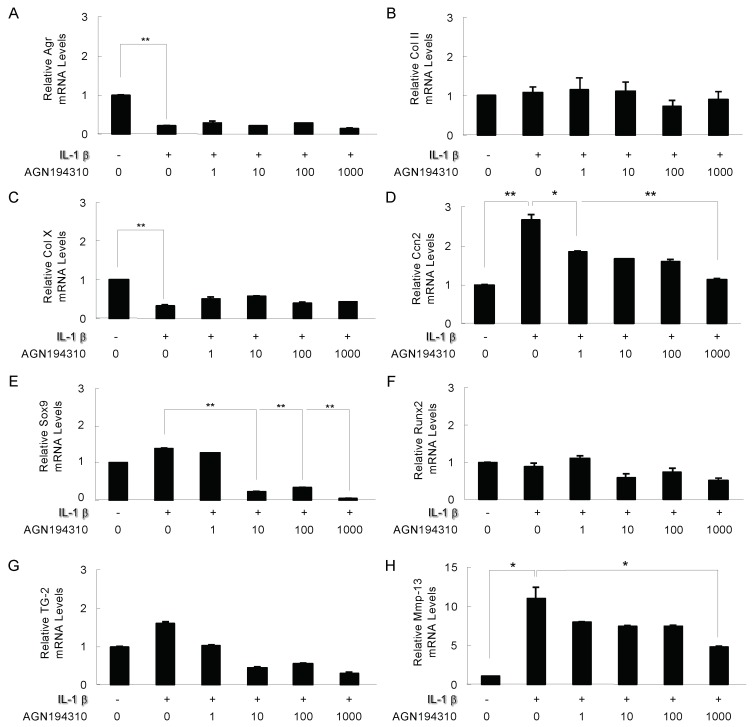
The effect of the RAR inverse agonist AGN194310 on IL-1β modulated genes. ATDC5 cells were exposed to 10 ng/mL IL-1β with the indicated amount of the RAR inverse agonist AGN194310 for 24 h. Total RNA from these cells was used in a real-time RT-PCR for the analysis of the mRNA expressions of Agr (**A**), Col II (**B**), Col10 (**C**), Ccn2 (**D**), Sox9 (**E**), Runx2 (**F**), Tg2 (**G**), and Mmp13 (**H**). The values in the graph indicate the relative mRNA level of the fold changes. Columns and S.D. bars, mean of triplicate determinations. Statistically significant differences (* *p* < 0.05, ** *p* < 0.01) between the indicated groups are marked by asterisks.

**Figure 4 ijms-21-02365-f004:**
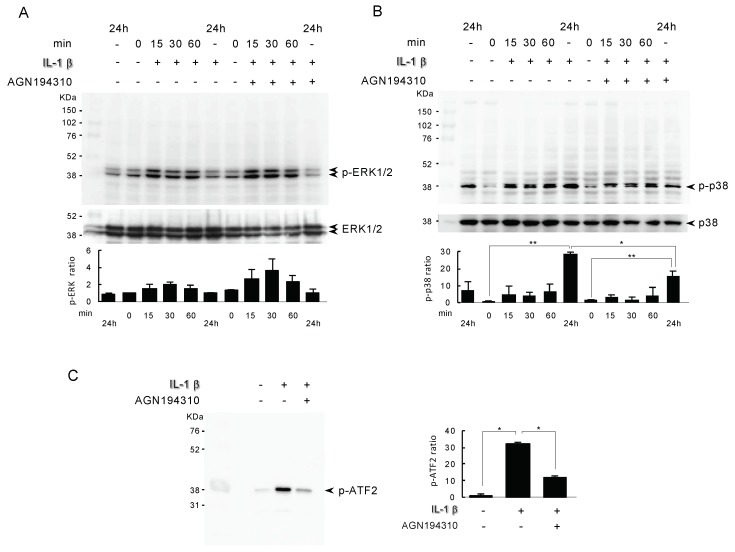
Analysis of the actions of the RAR inverse agonist AGN194310 on IL-1β-induced mitogen-activated protein kinase (MAPK). ATDC5 cells were exposed to 10 ng/mL IL-1β with/without 100 nM RAR inverse agonist AGN194310 for the indicated times, and the cell lysates were processed for an immunoblot determination of phosphorylated extracellular signal regulated-kinase (pERK1/2) (**A**) and p-p38 MAPK (**B**). (**C**) The level of activated p38 MAPK activity was determined by a kinase activity assay. Results shown are representative of three independent experiments, and further quantification by densitometry of triplicates. Densitometric analysis of pERK1/2 normalized to total ERK1/2 and p-p38 normalized to total p38, represented as times control. Bars show mean ± SD. Statistically significant differences (* *p* < 0.05, ** *p* < 0.01) between the indicated groups are marked by asterisks.

**Figure 5 ijms-21-02365-f005:**
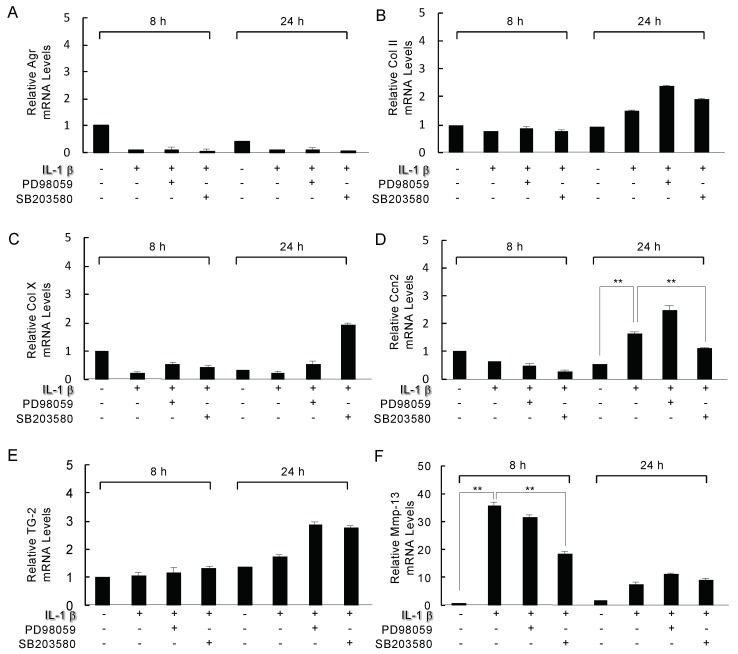
Analysis of the actions of the RAR inverse agonist AGN194310 and mitogen-activated protein kinase (MAPK) on effect of PD98059 or SB203580 on the IL-1β modulated genes. IL-1β-induced Agr, Col2, Col10, Ccn2 mRNA, Tg2, and Mmp13 mRNA. ATDC5 cells were exposed to 10 ng/mL IL-1β with/without 100 nM RAR inverse agonist AGN194310 for the indicated times, and the cell lysates were processed for an immunoblot determination of phosphorylated extracellular signal regulated-kinase (pERK1/2) and p-p38 MAPK (**A**), and phosphorylated activating transcription factor 2 (p-ATF2) (**B**). ERK1/2, p38 MAPK, and ATF2 was used as reference to quantify protein bands of pERK1/2 p-p38 MAPK and p-ATF2 by densitometry using ImageJ software individually. **C:** The level of activated p38 MAPK activity was determined by a kinase activity assay. ATDC5 cells were treated with 10 ng/mL IL-1β with or without 20 μM ERK1/2 kinase inhibitor PD98059 or p38 MAPK SB203580 for 24 h. Total RNA from these cells was used in a real-time RT-PCR analysis for aggrecan (**A**), Col II (**B**), Col10 (**C**), Ccn2 (**D**), Tg2 (**E**), and Mmp13 (**F**). The values in the graph indicate the relative mRNA level of the fold changes. The data from a typical experiment are presented; similar results were obtained in three separate experiments. Columns and S.D. bars, mean of triplicate determinations. Statistically significant differences (** *p* < 0.01) between the indicated groups are marked by asterisks.

**Figure 6 ijms-21-02365-f006:**
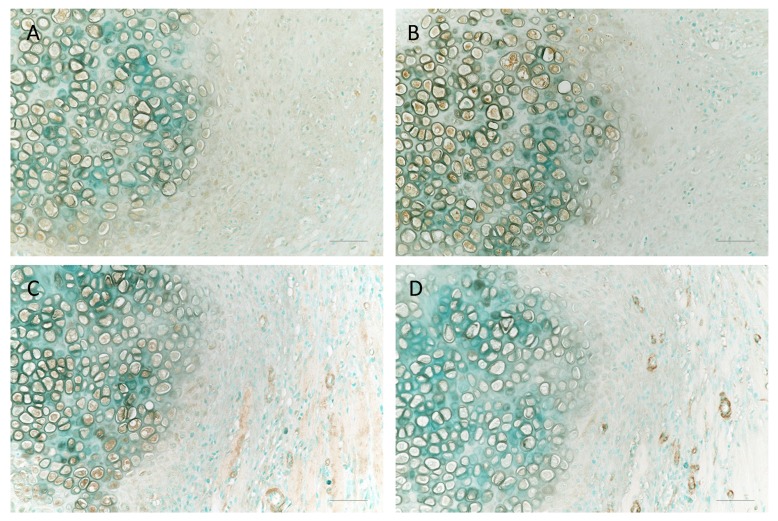
The localization of MMP-13, CCN2, p-p38 MAPK, and α-smooth muscle actin (α-SMA) in fractured rat ribs at 7 days post-fracture. Sections stained with MMP-13 (**A**), CCN2 (**B**), p-p38 MAPK (**C**) α-SMA (**D**), and methyl green counterstaining of cartilage at day 7 post-fracture. Each right photo showing a histological section is a magnification of the rectangle-delimited area in the corresponding left photo. Bar, 50 μm.

**Figure 7 ijms-21-02365-f007:**
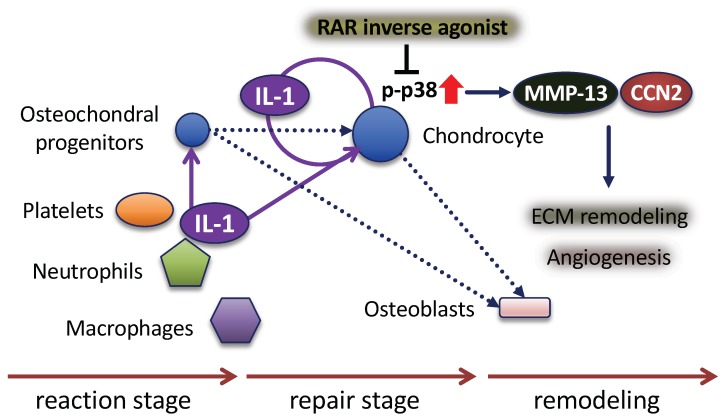
The expression and roles of IL-1β and RAR inverse agonists on endochondral bone ossification during fracture healing. Fracture healing can be viewed as a three-stage process. In the early fracture processes, platelets, neutrophils and macrophages secrete IL-1. Osteochondral progenitors differentiate into chondrocytes that proliferate to generate the early soft callus. Dotted line arrow: cell differentiation. Hypertrophic chondrocytes in the mature callus express angiogenic factors that result in vascular invasion. IL-1 induces the expressions of MMP-13 and CCN2 from chondrocytes via p38 MAPK, which is common with the RAR signal, suggesting that IL-1 plays an important role in ECM remodeling and angiogenesis (arrows).

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
