# Peer review of "Expression and Role of IL-1β Signaling in Chondrocytes Associated with Retinoid Signaling during Fracture Healing"

_ijms, 2020, doi:10.3390/ijms21072365_

Round 1

Reviewer 1 Report

The manuscript is interesting. However, there are several points that should be address by the authors.

In the statistical analysis paragraph of the methods the authors should report also the post doc analysis performed for multiple comparison by Anova.

Results should be separated from discussion in order to facilitate the reading of the manuscript.

No statistical analysis has been performed and reported in figure 2. The -/+ below the graphs should be aligned. The authors reported that they performed two independent experiments. The authors should perform at least three independent experiments. It should be briefly reported the role of the genes that the authors analyzed. Why the addition of all-trans-retinoic acid did not have a relevant effect?  

How many independent experiments have been reported in figure 3?

The authors reported that they performed two independent experiments in figure 4. The authors should do at least three independent experiments. Could the authors quantify the bands of the western blot and report a graph? The authors should also report the western blot of an housekeeping such as actin or tubulin.

Again no statistical analysis has been performed in figure 4 regarding gene expression analysis.

Figure 5: the authors should also report the staining performed (alcian blue?).

Author Response

REVIEWER #1:

  We greatly appreciate the reviewer’s insightful comments regarding our manuscript and are thankful for your considering our paper acceptable in accordance with the reviewers' comments. We have answered all of the reviewers’ comments below and densitometry analyzed, statistically reanalyzed and added the data in Figure 2, 3 and 4 accordingly and hope that you will now find the manuscript acceptable for publication in International Journal of Molecular Sciences.

Comment #1: In the statistical analysis paragraph of the methods the authors should report also the post doc analysis performed for multiple comparison by Anova.

Response #1: We have re-performed the statistical analysis based on all the data by using Tukey's test, Dunnett's test and Bonferroni correction for the analysis of multiple group comparisons. We have added the new analyzed data and text accordingly. We have added the name, Kyoichi Obata, as a co-author who has newly performed the statistical analysis at the title page and the Author Contributions.

Comment #2: Results should be separated from discussion in order to facilitate the reading of the manuscript.

Response #2: We have separated the Results from discussion and added each title accordingly. And we have newly added Conclusions. 

Comment #3: No statistical analysis has been performed and reported in figure 2. The -/+ below the graphs should be aligned. The authors reported that they performed two independent experiments. The authors should perform at least three independent experiments.

Response #3: We have re-performed the statistical analysis based on all the data by using Tukey's test. We have added the new analyzed data and text accordingly. We have aligned the description of the -/+ below the graphs accordingly.

The explanation of the dissertation was incorrect and has been revised as follows.

Two independent experiments were performed, and the average of the mRNA level is shown. The data from a typical experiment are presented; similar results were obtained in three separate experiments.

Comment #4: It should be briefly reported the role of the genes that the authors analyzed.

Response #4: We have added in the text about the role of the gene as described below.

“We next compared the effects of IL-1β on genes involved in endochondral bone formation with the effects of the RAR agonist on these genes described below. The aggrecan (Agr), the most abundant proteoglycan in the cartilage matrix, and type II collagen (Col II) for the markers of the chondrocytes, type X collagen (Col10), transglutaminase-2 (Tg2), Ccn2 and Mmp-13 for the markers of hypertrophic chondrocytes. Runx2 and Sox9 are the master transcription factors for bone and cartilage development respectively. Runx2 inhibit chondrocyte differentiation and Sox9 is essential for cell survival in differentiated chondrocyte lineage cells.”

Comment #5: Why the addition of all-trans-retinoic acid did not have a relevant effect?

Response #5: We have re-performed the statistical analysis including the data of the ATRA, and added the text, reference and statistical analysis data in the Figure 2 accordingly.

“ATRA increased the expression of Tg2 for 8 hr (**p<0.01) and 24 and 48 hr (*p<0.05) (Figure 2G). However, ATRA may be less specific inductive capacity than AGN204647 for ATDC5 cells the gene expression because of the pan-agonist of all the RARs [12].”

Comment #6: How many independent experiments have been reported in figure 3? The authors reported that they performed two independent experiments in figure 4. The authors should do at least three independent experiments.

Response #6: The explanation of the dissertation was incorrect and has been revised as follows in Figure 3 and 4.

Two independent experiments were performed, and the average of the mRNA level is shown. The data from a typical experiment are presented; similar results were obtained in three separate experiments.

Comment #7: Could the authors quantify the bands of the western blot and report a graph? The authors should also report the western blot of a housekeeping such as actin or tubulin.

Response #7: We have performed the quantify the bands of the western blot and added the text and the graph in Figure 3A and B accordingly. We have checked the ERK, p38 and ATF were not affected by RAR signal in the previous study [12], ERK1/2, p38 MAPK and ATF2 was used as reference to quantify protein bands of pERK1/2 p-p38 MAPK and p-ATF2 by densitometry using ImageJ software individually.

Comment #8: Figure 5: the authors should also report the staining performed (alcian blue?).

Response #8: We have added the information of staining in the Figure legend accordingly.

“methyl green counterstaining of cartilage”

Reviewer 2 Report

Tsuyoshi Shimo and colleagues performed a study with the aim to investigate the IL-1ß and retinoid acid receptor gamma (RARγ) signalling during fracture healing. For this purpose they used both in vivo and in vitro models.

The concept is of great interest and particularly important for its clinical implications. However, I think that this study need to be improved.

Major concerns:

Results and discussion:

I think that this section should be organized in a better way in order to present the obtained results more clearly. Separating results from discussion might be the best choose.

The results regarding the effects on Agr, Col II, Col X, CCN2, Sox9, Runx2, Tg2 and MMP-13 expression is presented only as relative mRNA levels. This is a limit of the study because it does not take into account the post-transcriptional gene regulation mechanisms. For this reason the effects of IL-1ß, RAR agonists and RAR invers agonists should be confirmed also at protein level.

Materials and methods:

“Real- time reverse transcription-polymerase chain reaction (RT-PCR)”

  • How cDNA has been obtained from total RNA? this information should be reported.
  • Which concentration of cDNA has been used for RT-PCR? this information should be reported.
  • More details about the method used for RT-PCR data analysis should be also included.
  • Representative Western blot bands of NF-kB protein should be included for all groups.

Minor concerns:

Cheek the text for typos i.e. “Mmp-13”, “CCN2” “Ccn2”.

Author Response

  We greatly appreciate the reviewer’s insightful comments regarding our manuscript and are thankful for your considering our paper acceptable in accordance with the reviewers' comments.  We have answered all of the reviewers’ comments below and modified the text and added references accordingly and hope that you will now find the manuscript acceptable for publication in International Journal of Molecular Sciences.

Results and discussion:

Comment #1: I think that this section should be organized in a better way in order to present the obtained results more clearly. Separating results from discussion might be the best choose.

Response #1: We have separated the Results from Discussion and added each title accordingly. And we have added Conclusions.

Comment #2: The results regarding the effects on Agr, Col II, Col X, CCN2, Sox9, Runx2, Tg2 and MMP-13 expression is presented only as relative mRNA levels. This is a limit of the study because it does not take into account the post- transcriptional gene regulation mechanisms. For this reason the effects of IL-1ß, RAR agonists and RAR invers agonists should be confirmed also at protein level.

Response #2: We have accepted the reviewer’s comments. We have realized the importance of the post-transcriptional gene regulation, we have added the text in Conclusion about the necessity of our future studies.

“However, further studies are needed with the post-transcriptional gene regulation and in vivo studies to clarify the role of RAR inverse agonists on the fracture healing.”

Materials and methods:

Comment #3:Real- time reverse transcription-polymerase chain reaction (RT-PCR)” How cDNA has been obtained from total RNA? this information should be reported. Which concentration of cDNA has been used for RT- PCR? this information should be reported. More details about the method used for RT-PCR data analysis should be also included.

Response #3: We have added the information in the text accordingly.

“Complementary DNA was generated from 1 μg of total RNA in a final volume of 20 μl by using a First Strand cDNA Synthesis Kit (Invitrogen Corporation, Grand Island, NY, USA). The real-time RT-PCR was performed with a Light Cycler (Roche Molecular Biochemicals, Mannheim, Germany) in Light Cycler capillaries with a commercially available master mix containing Taq DNA polymerase and SYBR-Green I deoxyribonucleoside triphosphate (Light Cycler DNA Master SYBR-Green I; Roche Molecular Biochemicals). After the addition of primers (final concentration: 10 μM), MgCl2 (3 mM), and template DNA to the master mix, 65 cycles of denaturation (95°C for 15 sec) and extension (60°C for 45 sec) were performed. After the PCR amplification was completed, a melting curve analysis was conducted. The fluorescence intensity of the double strand-specific SYBR Green I, reflecting the amount of formed PCR product, was monitored at the end of each elongation step. GAPDH mRNA levels were used to normalize the cDNA content of the samples.”

Comment #4: Representative Western blot bands of NF-kB protein should be included for all groups.

Response #4: Thank you for pointing out the NF-kB signal. We do not have data, but we have added the discussion and the references about the role of NF-kB signaling.

“We don't have data on the effect of AGN194310 on the activation of NF-κB involved in IL-1β signaling. ATRA reduced NF-κB activity and DNA binding [32], and p38 MAPK and NF-kB signaling have opposite biologic functions during inflammatory and osteogenic processes [33]. AGN194310 might regulate chondrocyte proliferation and differentiation to maintain endochondral ossification through accession of NF-κB and reduction of p38 MAPK signaling pathway [34]. However, the mechanisms of crosstalk between p38 MAPK and NF-kB signaling are not fully clarified [35], and require further investigation of AGN194310 effects on NF-kB signaling in future studies.”

New references                                         

  1. Rockel, S.J.; Kudirka, C.J.; Guzi, J.A.; Bernier, M.S., Regulation of Sox9 activity by crosstalk with nuclear factor-kappaB and retinoic acid receptors. Arthritis Res. Ther. 2008, 10, R3.
  2. Novack, D.V. Role of NF-kB in the skeleton. Cell Res. 2011, 21, 169-182.
  3. Nakatomi, C.; Nakatomi, M.; Matsubara, T.; Komori, T.; Doi-Inoue, T.; Ishimaru, N.; Weih, F.; Iwamoto, T.; Matsuda, M.; Kokabu, S.; Jimi, E. Constitutive activation of the alternative NF-κB pathway disturbs endochondral ossification. Bone. 2019, 121, 29-41.
  4. Chen, E.; Liu, G.; Zhou, X.; Zhang, W.; Wang, C.; Hu, D.; Xue, D.; Pan, Z. Concentration-dependent, dual roles of IL-10 in the osteogenesis of human BMSCs via P38/MAPK and NF-κB signaling pathways. FASEB J. 2018, 32, 4917-4929.

Minor concerns:

Comment #5: Cheek the text for typos i.e. “Mmp-13”, “CCN2” “Ccn2”.

Response #5: We have checked and corrected them according to the description of mRNA or protein.

Round 2

Reviewer 1 Report

The quality of the manuscript has been improved after the revision. I have only few comments.

Normally all the data obtained from the experiments (n=3) are entered in a software to obtain a graph reporting the error bars corresponding to the standard deviation obtained from the three experiments. Since the graphs reported in figures 2 and 3 report also error bars, it is not clear to me why the authors wrote “The data from a typical experiment are presented; similar results were obtained in three separate experiments.” in the legends of these figures. Do the error bars correspond to real-time PCR replicas of a single experiment? If so, the authors should graph all the data of all three experiments with the relative error bars corresponding to the standard deviation. In figure 4, the graph of real time experiments report no error bars.  

The western blot analysis quantification report no error bars. Again, the authors should quantify the protein expression using Image J software in the three experiments and report a graph with error bars and statistical analysis. The use of ERK 1/2, p38 and ATF as reference genes is very questionable. Western blot analysis is normally performed using an housekeeping gene (i.e. tubulin, GADPH or actin).

Author Response

REVIEWER #1:

  We greatly appreciate the reviewer’s insightful comments regarding our manuscript and are thankful for your considering our paper acceptable in accordance with the reviewers' comments. We have answered all of the reviewers’ comments below and densitometry analyzed, statistically reanalyzed and added the data in Figure 4 accordingly.

Comment #1: Normally all the data obtained from the experiments (n=3) are entered in a software to obtain a graph reporting the error bars corresponding to the standard deviation obtained from the three experiments. Since the graphs reported in figures 2 and 3 report also error bars, it is not clear to me why the authors wrote “The data from a typical experiment are presented; similar results were obtained in three separate experiments.” in the legends of these figures. Do the error bars correspond to real-time PCR replicas of a single experiment? If so, the authors should graph all the data of all three experiments with the relative error bars corresponding to the standard deviation. 

Response #1: As pointed out by the reviewer, we have graphed all the data of all three experiments with the relative error bars corresponding to the standard deviation. The text was corrected according to the statistical analysis data of all three experiments.

Comment #2: In figure 4, the graph of real time experiments report no error bars.

Response #2: As pointed out by the reviewer, we have graphed all the data of all three experiments with the relative error bars corresponding to the standard deviation.

Comment #3: The western blot analysis quantification report no error bars. Again, the authors should quantify the protein expression using Image J software in the three experiments and report a graph with error bars and statistical analysis.

The use of ERK 1/2, p38 and ATF as reference genes is very questionable. Western blot analysis is normally performed using an housekeeping gene (i.e. tubulin, GADPH or actin).

Response #3: We have quantified the protein expression using Image J software in the three independent experiments and report a graph with error bars and statistical analysis in Figure 4. As pointed out by the reviewer, we have performed western blotting by using actin (goat IgG) antibody from Santa Cruz Biotechnology (Santa Cruz, CA) at a 1:200 dilution as we described previously (Shimo, T.et al. PLoS One 2016, 11, e0151731.). However, the actin bands were not detected clearly because of the problems of protein degradation. In this study, we have subjected a 50-μg sample of each lysate to SDS-PAGE, and checked reliable detection of protein bands on transferred nylon membranes with 0.1% Ponceau S in 5% acetic acid. We have shown the densitometric analysis using pERK1/ 2 normalized to total ERK1/2 and p-p38 normalized to total p38 [PLoS One. 8 (6): e65150. 2013].

Reviewer 2 Report

Major concerns

Figure 4

Authors wrote:  “ERK1/2, p38 MAPK and ATF2 was used as reference to quantify protein bands of pERK1/2 p-p38 MAPK and p-ATF2 by densitometry using ImageJ software individually”.

  • What means this sentence? Reference is an endogenous control i.e. glyceraldehyde-3-phosphate dehydrogenase (GAPDH) and beta-actin (ACTB). Please clarify.
  • Western blot bands of reference protein used should be reported in the figure above the western blot bands of the target proteins.
  • Molecular weight indicator should be also included in the western blot bands .

Material and methods

4.4 Immunoblot analysis

  • The reference protein used to quantify protein expression should be indicated.
  • Please give more details about the primary and secondary antibodies used.

4.6 Real- time reverse transcription-polimerase chain reaction (RT-PCR)

  • How gene expression analysis was performed? Please give more details about the methods used.
  • Authors wrote :“After the addition of primers (final concentration: 10 μM), MgCl2 (3 mM), and template DNA to the master mix, 65 cycles of denaturation (95°C for 15 sec) and extension (60°C for 45 sec) were performed.” Why 65 cycles? It is a very strange value in a qPCR protocol. Please clarify.

Author Response

REVIEWER #2:

  We greatly appreciate the reviewer’s insightful comments regarding our manuscript and are thankful for your considering our paper acceptable in accordance with the reviewers' comments.  We have answered all of the reviewers’ comments below and modified the text and added reference accordingly.

Comment #1: Figure 4 Authors wrote:  “ERK1/2, p38 MAPK and ATF2 was used as reference to quantify protein bands of pERK1/2 p-p38 MAPK and p-ATF2 by densitometry using ImageJ software individually”. What means this sentence? Reference is an endogenous control i.e. glyceraldehyde-3-phosphate dehydrogenase (GAPDH) and beta-actin (ACTB). Please clarify.

Western blot bands of reference protein used should be reported in the figure above the western blot bands of the target proteins.

Molecular weight indicator should be also included in the western blot bands.

Response #1: We have quantified the protein expression using Image J software in the three independent experiments and report a graph with error bars and statistical analysis in Figure 4. As pointed out by the reviewer, we have performed Immunoblot analysis by using actin (goat IgG) antibody from Santa Cruz Biotechnology (Santa Cruz, CA) at a 1:200 dilution as we described previously (Shimo, T.et al. PLoS One 2016, 11, e0151731.). However, the actin bands were not detected clearly because of the problems of protein degradation. In this study, we have subjected a 50-μg sample of each lysate to SDS-PAGE, and confirmed reliable detection of protein bands on transferred nylon membranes with 0.1% Ponceau S in 5% acetic acid. We have shown the densitometric analysis using pERK1/2 normalized to total ERK1/2 and p-p38 normalized to total p38 [PLoS One. 8 (6): e65150. 2013].

We have replaced the photo with Immunoblot overall images in Figure 5A-C.

The molecular weight indicator which was loaded in the left lane in each data is also shown.

Comment #2: 4.4 Immunoblot analysis

The reference protein used to quantify protein expression should be indicated.

Please give more details about the primary and secondary antibodies used.

Response #2:

As pointed out by the reviewer, we have replaced the photo with Immunoblot overall image in Figure 5A-C. The molecular weight indicator which was loaded in the left lane in each data is also shown.

We have added details about the primary and secondary antibodies used in the text.

Comment #3: 4.6 Real- time reverse transcription-polimerase chain reaction (RT-PCR)

How gene expression analysis was performed? Please give more details about the methods used.

Authors wrote :“After the addition of primers (final concentration: 10 μM), MgCl2 (3 mM), and template DNA to the master mix, 65 cycles of denaturation (95°C for 15 sec) and extension (60°C for 45 sec) were performed.” Why 65 cycles? It is a very strange value in a qPCR protocol. Please clarify.

Response #3: We have confirmed our data and described the methods in the text. We have newly added our previous study based on the conditions of the analysis.

49.  Shimo, T.; Matsumoto, K.; Takabatake, K.; Aoyama, E.; Takebe, Y.; Ibaragi, S.; Okui, T.; Kurio, N.; Takada, H.; Obata, K.; Pang, P.; Iwamoto, M.; Nagatsuka, H.; Sasaki, A. The Role of Sonic Hedgehog Signaling in Osteoclastogenesis and Jaw Bone Destruction. PLoS One 2016, 11, e0151731.

Round 3

Reviewer 2 Report

The quality of the manuscript has been improved and there are no other points to be addressed.